# Surface Optimization of Commercial Porous Ti Substrates by EPD of Titanium Nitride

**DOI:** 10.3390/membranes12050531

**Published:** 2022-05-19

**Authors:** Cecilia Mortalò, Maria Cannio, Valentina Zin, Enrico Miorin, Francesco Montagner, Luca Pasquali, Monica Montecchi, Dino Norberto Boccaccini, Monica Fabrizio, Silvia Maria Deambrosis

**Affiliations:** 1National Research Council of Italy-CNR, Institute of Condensed Matter Chemistry and Technologies for Energy–ICMATE, Corso Stati Uniti 4, 35127 Padova, Italy; cecilia.mortalo@cnr.it (C.M.); enrico.miorin@cnr.it (E.M.); francesco.montagner@cnr.it (F.M.); silviamaria.deambrosis@cnr.it (S.M.D.); 2Department of Engineering “Enzo Ferrari”, University of Modena and Reggio Emilia, Via Pietro Vivarelli 10, 41125 Modena, Italy; luca.pasquali@unimore.it (L.P.); monica.montecchi@unimore.it (M.M.); 3IOM-CNR, Strada Statale 14, Km. 163.5 in AREA Science Park, Basovizza, 34149 Trieste, Italy; 4Department of Physics, University of Johannesburg, P.O. Box 524, Auckland Park 2006, South Africa; 5Tecno Italia Digital Via Emilia Romagna 83, 41049 Sassuolo, Italy; dinoboccaccini@hotmail.com; 6CNR Engineering ICT and Technologies for Energy and Transportation Department National Research Council of Italy, Piazzale Aldo Moro 7, 00185 Rome, Italy; monica.fabrizio@cnr.it

**Keywords:** porous Ti substrates, *multi-step* EPD, titanium nitride infiltration

## Abstract

In this work, the infiltration of TiN powders by electrophoretic deposition (EPD) in aqueous media was considered as alternative method to reduce the size craters and the roughness of commercial porous Ti substrates. Ti substrates can be used as suitable supports for the deposition of dense hydrogen separation TiNx-based membranes by physical vapor deposition (PVD) techniques. The influence of various EPD deposition parameters on surface morphology and roughness of TiN-infiltrated substrates were investigated in order to optimize their surface properties. The results suggest that a *multi-step* EPD procedure is an effective technique for reducing substrate surface defects of commercial porous Ti substrates which could then be successfully used as proper supports for the deposition of dense and defect-free TiNx layers, also aligning the thermal mismatch between the active layer and the porous substrate.

## 1. Introduction

In the last years, hydrogen has been considered as a promising energy vector for various industrial applications [1,2]. However, the sustainability of H_2_ as an energy carrier and in other important technological sectors is deeply linked to the hydrogen separation and purification technologies. In this context, dense metallic membranes based on Pd and its alloys have been extensively proposed due to their high theoretical permeability, infinite selectivity, and thermo-mechanical stability under operational conditions [3,4,5,6,7]. However, because of the high cost of palladium [8], alternative materials are strongly demanded and investigated. In this regard, dense ceramic membranes based on mixed protonic-electronic conductors (MPECs) represent potential materials for hydrogen purification, especially at high temperatures (typically higher than 600 °C) [9,10,11]. Among this kind of materials, ceramic–ceramic composites based on the perovskite-fluorite BaCe_1-x_Zr_x_Y_y_O_3-δ_-doped CeO_2_ systems have attained considerable attention owing to their chemical stability and mechanical robustness under reducing and acid-containing atmospheres, such as CO_2_ and H_2_S [12,13,14,15]. On the other hand, the possibility to use dense ceramic membranes based on some metal nitrides for hydrogen separation at intermediate temperature (25 °C to 500 °C) have also been investigated. Kim et al., prepared TiN-based composite membranes by spark plasma sintering [16] and hot press sintering [17,18], showing high hydrogen permeability values in the 200 to 500 °C temperature range. Nozaki et al., demonstrated the effectiveness of HfN layers to improve the high temperature stability of Pd membranes on Ta substrates. These HfN layers displayed an adequate hydrogen permeability at 300 °C [19]. More recently, selective hydrogen permeability through TiN and HfN thin films deposited on Al_2_O_3_ porous ceramic supports by radio frequency (RF) reactive sputtering was observed by Kura et al. [20,21]. The authors suggest that the hydrogen permeability in these materials is based on hydride ion formation and electron conduction. In Ref. [22] Saito et al., demonstrated that hydride ions defects are also incorporated in *n*-type semiconductor ο-Zr_3_N_4-δ_ thin films under hydrogen atmosphere. Despite the impressive performance of these materials in the RT to 500 °C temperature range, the synthesis of dense, defect-free nanocrystalline films on porous supports is challenging and makes it difficult to assess quantitatively the hydrogen separation phenomena. Indeed, an optimal fine control of the grown layer’s stoichiometry and microstructure is crucial for the effective application of these materials as hydrogen separation membranes under operational conditions. Among different available technologies for depositing thin coatings, reactive sputter deposition offers the opportunity to tailor the film properties [23,24,25,26,27]. Recently, Mortalò et al., demonstrated the effectiveness of the reactive high power impulse magnetron sputtering (RHiPIMS) technique for the deposition of dense nanocrystalline TiN_x_ (x = 0.4–1.0) films [28]. Coatings having 500 nm thickness were successfully deposited on Si, exhibiting good adhesion even after a heat treatment at 500 °C in hydrogen-containing atmosphere. However, the films grown on porous alumina displayed poorer adhesion especially for coating thicknesses higher than 500 nm. This behavior could be ascribed to the high level of residual stresses in coatings deposited on alumina substrates, considering the deposition conditions: temperature of 300 °C and thermal expansion coefficients of α_f_ = 7.2 × 10^−6^ K^−1^ for TiN coating [29] and α_s_ = 12.662 × 10^−6^ K^−1^ for Al_2_O_3_ [30] respectively. Therefore, to avoid the occurrence of residual stresses, other suitable porous supports must be investigated for the preparation of TiN-based dense ceramic membranes for hydrogen separation. Porous metal supports are promising alternatives to ceramics due to their superior mechanical strength, resistance against cracking, ease of module fabrication and sealing. Indeed, commercial stainless steel, nickel, Inconel, and some Ti-based alloys are considered as suitable porous supports and are currently investigated [31,32,33]. Taking into consideration the good thermal and mechanical stability of titanium at high temperature and under harsh and corrosive conditions [32,33,34,35,36] and its thermal expansion coefficient matching with that of TiN coatings (i.e., α_s_ = 8.5 × 10^−6^ K^−1^ [37]; α_TiN_ =7.2 × 10^−6^ K^−1^ [29]), commercial porous Ti substrates were investigated in this work. However, these metallic supports are usually characterized by relatively irregular large pores with a wide pore size distribution and very rough surfaces, which hinder the PVD deposition of thin and defect-free coatings. Therefore, it is crucial to optimize the substrate surface prior to the deposition process because the coating/substrate adhesion plays a key role in improving the mechanical and functional properties of the film. To this end, in addition to conventional cleaning procedures, some surface treatments, such as (I) mechanical treatments, (II) chemical treatments, and (III) incorporation of an intermediate layer, may be performed to increase the film adherence and/or reduce the pore sizes and roughness of the supports’ surface. Depending on the final applications, different techniques have been explored, such as chemical methods, laser structuring technique, laser nitriding process, etc., [38,39,40,41]. Another method that could be explored to reduce the roughness and the presence of defects on a porous surface is viscous fingering [42]. However, most methods require expensive equipment and/or the use of hazardous chemicals and/or processes. Hence, the look for non-expensive and more environmental ecofriendly methods is currently of great interest in the scientific community [31]. To this purpose, the infiltration of TiN powders by electrophoretic deposition (EPD) was investigated to optimize both pore size distribution and surface roughness of Ti substrates. TiN powders were selected for EPD infiltration to enhance the adhesion of active TiN layer deposited by HiPIMS. The EPD technique is a colloidal process, wherein charged particles in a stable suspension move under an applied electrical field, thus alloying the deposition of different type of coatings and infiltration of dense or porous substrates with micro- and/or nano-powders. EPD is generally carried out at room temperature and standard pressure. A wide range of structures with different composition and complex shapes can be obtained with traditional and/or advanced materials to find applications in different fields, e.g., multilayered composites, solar cells, batteries and electrochemical capacitors, solid oxide fuel cells, sensors, biomedical materials, catalysts, and many others. Compared to other material processing methods, EPD has the advantages of flexibility, simple apparatus and equipment, short processing time and cost-effectiveness [43,44,45,46,47,48,49,50,51,52]. Despite the simplicity of the technique, very few studies report the deposition of TiN coatings by EPD and all are devoted to the preparation of protective films on dense substrates. Among these, some works are focused on the deposition of TiN coatings on dense stainless steel as protective corrosion layer for bipolar plates of polymer electrolyte fuel cells [53,54,55,56,57]. A couple of papers report the EPD deposition of TiN coatings on dense Ti substrates. For example, Cui et al., describe the preparation of homogeneous and adhesive nanocrystalline TiN thin films from aqueous suspensions of TiN nanometric powders, obtaining 6 µm thick coatings after 10 min of deposition time by applying 10 V of deposition voltage [58]. More recently, Ureña [59] and Mendoza [60] prepared TiN protective coatings on Ti substrates to improve their wear resistance and to reduce the surface Young’s modulus for biomedical applications. Homogeneous coatings were obtained after 5 min by applying 90 V of deposition voltage. Only one paper reports the deposition of TiN powders on porous substrates, i.e., Ni foams [61]. Successful EPD material processing requires the preparation of stable suspensions and the adjustment of process parameters such as the applied voltage and time. Suspension properties and stability depend on the physiochemical nature, size, and concentration of the suspended particles, chemical nature of the solvent, as well as type and concentration of both the dispersants and the additives. Most of the papers in the literature report the study of suspensions of TiN powders in organic solvents. Mendoza et al. [57,60] and Ureña et al. [59] prepared TiN nanopowder suspensions using isopropyl alcohol as solvent and polyethilenimine (PEI) as stabilizer. Isopropanol was also used as media for the preparation of TiN-based suspensions on stainless steel substrates by EPD [53,55]. Kavanlouei deposited TiN coatings on stainless steel discs by using butanol as solvents and tri-ethanolamine as dispersant agent [56]. Lee used suspensions in anhydrous ethanol in the presence of PEI and poly(diallyldimethylammonium chloride) as stabilizers [54]. Some works report the stabilization of aqueous titanium nitride suspensions. Shih et al., examined the stabilization of aqueous titanium nitride suspensions with ammonium salt of poly (methacrylic acid) at various pH values [62]. Zhang et al. [63] and Z. Gonzales et al. [61] investigated the stability of TiN powder suspensions in aqueous media with PEI as dispersion surfactant, while Guo et al., used Tween 80 (polyoxyethylene 80 sorbitan monooleate) [64]. Only one paper reports the EPD deposition of TiN coatings from aqueous suspensions of nanometric powders without using organic dispersants [58]. The use of aqueous suspension for EPD deposition is challenging, due to the electrolysis of water, since evolution of hydrogen and oxygen gases occurs at the electrodes, affecting the final quality of the deposits [45]. However, aqueous EPD is attractive for the application in industrial scale processes as it has a lower environmental impact than organic media, it is safe and cost-effective. Moreover, the deposition rates in organic solvents are significantly lower in comparison to those obtained in aqueous suspensions because of the much lower permittivity. Thus, very high deposition voltages are required to achieve acceptable shaping time [65]. In this work, aqueous suspensions were investigated for the first time for the electrophoretic impregnation of TiN powders on porous Ti substrates. A home-made EPD cell was specifically designed to allow the infiltration of TiN powders only on one side of the substrate. Moreover, different experimental parameters have been considered, such as the applied voltage, deposition time, and different grain size dimensions of TiN powders. Lastly, the effectiveness of a multi-step procedure with respect to the single-step deposition for the reduction of crater defects and surface roughness was demonstrated.

## 2. Materials and Methods

### 2.1. Starting Materials

Porous Ti discs (28–30% of porosity, declared average pore size 0.4 μm) with 2 mm of thickness and 20 mm diameter were purchased by Edgetch Industries LLC (Tamarac, FL, USA). The surface of porous commercial Ti supports was polished by using abrasive SiC papers with European *p* grade from 400 to 2500 (REMET) and ultrasonically cleaned in acetone (Sigma Aldrich, ACS reagent, ≥99.5%) and absolute ethanol (Sigma Aldrich Merk Life Science S.r.l., Milan, Italy, ≥99.8%) for 30 min sequentially. Micro-sized (99.9%, APS 1 μm) and nano-sized titanium nitride powders (99.9%, APS 30 nm), designated as TiN-micro and TiN-nano respectively, were supplied by Edgetch Industries LLC (Tamarac, FL, USA) and used as raw materials in this research to prepare suitable suspensions for EPD deposition experiments.

### 2.2. Preparation of TiN Suspensions

TiN-micro, TiN-nano, and bimodal TiN-micro and –nano (weight ratio 1:1) powder suspensions were prepared in water without any organic dispersants. All suspensions used for EPD experiments were prepared by dispersing 1 wt.% of TiN powders and magnetically stirred for 30 min. Then the suspensions were sonicated with a probe sonicator (ULC 400, 36 KHz, WEBER ULTRASONICS AG, 76307 Karlsbad, Germany) for 30 min to break agglomerates. Fresh suspensions were used for each EPD process.

### 2.3. TiN Powders Infiltration by EPD

EPD experiments were performed at room temperature using a two-electrode cell consisting of a porous Ti disc having 20 mm of diameter and 2 mm of thickness, acting as the substrate material, and a AISI 316L stainless steel spiral of similar surface area used as counter electrode. The *home-made* electrochemical cell used for the EPD experiments, consists in: (I) A PET tank containing the suspension to be deposited, (II) a stainless-steel cathode, and (III) a potentiostat/galvanostat (Versastat3–400, potential range ±12 V, Princeton Applied Research, 801 South Illinois Avenue, Oak Ridge, TN, USA). The anode is made up of porous Ti substrates. In Figure 1, the flow chart of the EPD deposition process is shown.

TiN powders were infiltrated on polished porous Ti discs by applying different direct current (DC) deposition potentials: 10 V for suspensions containing micrometric powders (TiN-micro and TiN-nano/micro) and 5 V for the suspension containing only nanometric powders (TiN-nano). The distance between the electrodes was set at 1 cm.

Table 1 summarizes the experiments carried out in this investigation. In particular, *single-step* and *multi-step* procedures were considered for the EPD tests.

#### Single Steps Tests

The single-step method consisted in a continuous deposition of TiN powders at the constant voltage values specified in Table 1. Preliminary experiments performed at different times in the range from 600 s up to 3600 s had been already carried out. Table 1 reports a selection of the tests with the most outstanding results achieved so far, as it is discussed in Section 3.

After each deposition, the excess of TiN powders placed on the surface by gravity was removed by washing with distilled water. Then, the fresh deposits were dried in air for 24 h at room temperature.

### 2.4. Multi-Step Tests

In line with the single-step mode procedure, the experiments using multi-step processes (EPD TiN-06, EPD TiN-07, EPD TiN-08 and EPD TiN-09), were carried out at a voltage of 5 V with suspensions of TiN-nano powders, while a potential of 10 V was used with the suspensions containing micrometric powders. The experiments were conducted in subsequent stages with variable infiltration time according to the particle size of the suspended powders but covering a total deposition time of maximum 3600 s, as shown in Table 1. After each experiment, the excess of TiN powders on the surface was removed by washing with distilled water and the suspension was stirred with an ultrasonic probe for 15 min before the next deposition step. The final fresh deposits were air dried for 24 h at room temperature in all the performed experiments.

In EPD TiN-06 experiment, a procedure based on 20 steps of 60 s each was applied for the blended suspensions of TiN-micro or TiN-nano/micro powders at 10 V, followed by four deposition steps at 5 V of 300 s each with suspensions containing only TiN nanometric powders. In the experiment named EPD TiN-07, TiN-nano/micro powders were deposited following a procedure consisting in 30 deposition steps of 60 s each. Then, 4 steps of 300 s were used for the suspensions containing only nanometric powders. EPD TiN-08 TiN was obtained from TiN-micro powders deposited following a procedure consisting in 20 deposition steps of 60 s each. Finally, 4 steps of 300 s were used for the suspensions containing only nanometric powders. In the EPD TiN-09 multi steps experiment, 8 steps of 300 s each at 5 VV were carried out using only nano TiN powders to infiltrate Ti porous substrates.

### 2.5. Characterization

The morphological characterization of TiN powders and porous Ti substrates was performed by scanning electron microscopy (Nova NanoSEM 450, FEI Company, 5350 NE Dawson Creek Drive, Hillsboro, OR, USA). SEM analyses were also performed to investigate the penetration depth of the TiN powders into the porous Ti substrates and coatings quality. Crystal structure and phase purity information of TiN powders and Ti substrates were obtained by XRD analyses by using multi-purpose X’Pert PRO (PANalytical, Almelo, The Netherlands) diffractometer equipped with a Cu Kα source operating at 40 kV and 40 mA. Powder X-ray diffraction patterns were collected at room temperature using a step scan procedure (0.002°/2θ step, 5 s time per step) in the 20–100° 2θ range. The surface layers’ composition of supplied TiN-nano and TiN-micro powders was determined by X-ray photoelectron spectroscopy (XPS). For XPS measurements, Mg Kα radiation was used from a dual anode non-monochromatic X-ray source (VG-XR3) operated at 15 mA, 15 kV. Spectra were acquired with a VG CLAM2 hemispherical analyzer (VG-Microtech) at constant pass energy of 25 eV. Specific surface area of TiN micro and nano commercial powders were estimated by the Brunauer–Emmett–Teller (BET) method (Micromeritics Gemini VII 2390 Series). Zeta potential of as-prepared TiN suspensions were measured by laser Doppler velocimetry using a Zeta Sizer ZS analyzer (Malvern Zeta Sizer ZS ZEN 3600, Malvern Panalytical L.t.d., Malvern, UK). For the zeta potential measurements, suspensions with 0.1 wt.% of TiN powders were prepared analogously as described before. Stylus contact profilometer (Dektat XT, Bruker, Billerica, MA, USA) was used to analyze the surface roughness and topology of the Ti substrates before and after the EPD treatments. Optical microscopy images were collected by a stereomicroscope at a magnification of 40x (Leica/Wild M3Z Stereo Microscope, Leica Microsystems Srl, Buccinasco, Milan, Italy) and image analysis was carried out by means of Image J software v1.46 on optical microscopy pictures to evaluate the overall residual porosity after the EPD treatments. Generally, the Feret‘s statistical diameter is used to measure the size of irregularly shaped particles (or holes) and it is determined for randomly oriented particles (or holes), thus giving an average value over all possible orientations [66]. Hence, it has been used in this study to compare the surface state of Ti substrate and differently treated surfaces. Different roughness parameters have been investigated to describe better the surface topology prior and after EPD treatments in different conditions [67]. The different quantities relating to the surface roughness are Ra, Rq, Rku, and Rsk. The roughness Ra is defined as the arithmetic mean value of the deviations (taken as an absolute value) of the real profile of the surface with respect to the mean line; Rq is the quadratic mean of the deviations of the profile points from the mean line. The roughness Rsk (Skewness) is the measure of the symmetry of the profile with respect to the mean line, i.e., the measure of the average of the first derivative of the surface (the departure of the surface from symmetry). If the graph is symmetrical with respect to the mean line (normal distribution) the Rsk value is equal to 0. On the other hand, if a deviation below or above the mean line is observed, Rsk < 0 or Rsk > 0 are obtained respectively. Finally, the roughness Rku (Kurtosis) is a measure of the sharpness of the surface height distribution within the sampling length and characterizes the spread of the height distribution. A surface with a Gaussian height distribution has a kurtosis value of three. If Rku > 3 there is a distribution of sharp, sharp peaks and valleys along the sampling length, and a Rku < 3 suggests a uniform distribution of peaks and valleys over the sampling length.

## 3. Results and Discussion

### 3.1. Characterization of TiN Powders

SEM images of as-received micro- and nano-TiN powders are depicted in Figure 2a,b respectively.

As shown in the Figure 2, micro-TiN powders exhibit irregular particle shape with board size distribution raging from submicron to 20 μm, while nano-TiN powders show a more homogenous grain distribution with dimension in the 30–50 nm range. The specific surface areas of as-received micro-TiN and nano-TiN powders are 2.38 m^2^/g and 33.24 m^2^/g respectively. Figure 3 shows XRD patters of as-received micro-size and nano-size TiN powders.

XRD profiles of both TiN samples show reflections which are assigned all to the (111), (200), (220), (311), (222), and (400) crystal planes corresponding to a *fm-3m* face centered cubic phase, (space group 225, JCPDS 38–1420) [56]. No other peaks related to other secondary phases are present in the spectra. In Figure 4, XPS photoemission spectra of TiN-micro and TiN-nano powders are displayed. The XPS core level spectra for Ti 2p and *n* 1s are shown in Figure 4a,b respectively. In the deconvoluted Ti 2p spectra, the different contributions of titanium in the system are shown, represented by different Voigt-type doublets due to the spin-orbit interaction (2p3/2–2p1/2 spin orbit splitting of about 6.4 eV). In particular, the doublet with the Ti 2p3/2 peak centered at about 456 eV is associated to the presence of TiN. The doublet with 2p3/2 component at about 459 eV is related to the presence of oxy-nitride structures, TiOxNy, in good agreement to data reported in the literature [28,63,64,68]. In Figure 4b, the *n* 1s spectra were de-convoluted with two peaks, the main at 397 eV and the smaller at 399 eV. The peak at lower binding energy is ascribable to TiN, while the peaks at about 399 eV is associated to oxy-nitride phases, TiNxOy [68,69]. To conclude, the Ti 2p and *n* 1s spectra suggest the presence of TiN and TiNxOy on the surface of as-received micro- and nano-TiN powders suggesting the spontaneous surface oxidation of TiN powders due to their exposure to the atmosphere. As observable, there is a progressive relative increase of the component linked to titanium in the nitride as the particle size increases. This is due to the higher surface area of nanosized powders which are more sensitive to the oxidation.

### 3.2. Characterization of As-Received and Polished Ti Substrates

From SEM micrographs (not reported here) of as-received commercial porous substrates, irregular large pores with a wide pore size distribution are evident: indeed, open porosity with a pore distribution varying in the submicron-50 μm size range is evident. On the other hand, the mechanical polishing treatment successfully improves the pore size spreading and surface roughness of the Ti substrate. However, the presence of a few large craters (with dimensions > 10 µm) is still evident. As measured by optical microscopy on Ti treated surface, the porosity size distribution is centered at 40 µm of Feret’s diameter. Considering their potential as porous substrates for the deposition of TiN coatings by HiPIMS technique, the high porosity size distribution could be detrimental to obtain dense and defect-free layers. Hence, the electrophoretic infiltration of TiN powders becomes crucial to further improve the external surface of the porous Ti supports.

### 3.3. Characterization of TiN Suspensions

Considering the pore size in the polished Ti substrates ranging from the nanometric to micrometric values, in this study micro- and nano-sized TiN powders were selected for the EPD infiltration. The stability of the suspensions was evaluated by measuring the potential ζ, in accordance with the “DLVO” theory, based on the potential energy curve of the dispersed particles as a function of their distance [43,70,71,72]. According to this theory, the stability of a colloidal suspension depends on the balance of repulsive and attractive forces existing among the suspended particles. A high potential value ζ indicates greater stability of colloidal systems since electrostatic repulsions are generated, preventing the particles aggregation. Generally, systems with potential values ζ > 30 mV and ζ < −30 mV are considered stable. Table 2 reports the ζ potential values measured for all prepared suspensions. The ζ potential of all the suspensions measured under neutral conditions assumes negative values, thus indicating the accumulation of positive charges near the particles surface and, therefore, their electrically negative nature.

Data are in good agreement with results reported in literature. Guo et al. [64] and Zhang et al. [63] measured ζ potential values of aqueous suspensions of TiN nanometric and micrometric powders without dispersants of about −20 mV and –60 mV at pH = 7 respectively. In this case, the absolute value of negative zeta-potential increases with reducing the particle size: indeed, the suspension of the nano-sized powders shows higher value than that obtained with the suspension based on TiN micro powders. Moreover, the potential ζ value of TiN nano/micro suspension is intermediate. Generally, there is an inverse relationship between size and zeta potential of the nanoparticles. Since the particles with small diameter are easily affected by the random movement of fluid flow and other particles, their absolute value of effective zeta-potential is expected to be greater than that of larger particles [72]. Moreover, ζ values are influenced also by dispersion conditions. In this work, dispersions are prepared by an initially stirring followed by an ultrasonic probe treatment. The different dispersion conditions could explain the higher ζ values obtained for these TiN-nano suspensions compared to those reported by Zhang et al., and Guo et al. [63,64]. Lastly, the stability of the TiN aqueous suspensions over time was also evaluated. Figure 5 shows the trend of the ζ potential over time of the suspensions of TiN-nano, TiN-micro, and TiN-nano/micro powders in H_2_O.

The graph shows that the ζ potential of all three suspensions decreases slightly in absolute value over time, indicating a slight deterioration in stability. However, after 300 min, the ζ values still fall within the stability range, especially for TiN-nano and TiN-nano/micro suspensions. Indeed, colloidal particles, which are 1 µm or less in diameter, tend to remain in suspension for long periods due to the Brownian motions. On the other hand, particles larger than 1 µm require continuous hydrodynamic agitation to remain in suspension [43]. Therefore, it can be concluded that all the suspensions of TiN powders in water show adequate stability and are suitable for the deposition process by EPD, without the addition of any organic dispersant.

### 3.4. TiN Powders Infiltration by EPD 

EPD infiltration of TiN powders has the purpose of reducing the average size of the roughness and the pore size to obtain a suitable substrate for the subsequent PVD deposition process. Moreover, EPD could also improve the thermal mismatch and adhesion between the porous substrate and the active TiN coating of the asymmetrical membrane. To this aim, TiN infiltration experiments by EPD were conducted after the polishing treatment. Table 1, reported in Section 2.3, summarizes the deposition experiments that have given the most satisfactory results. In particular, the influence of different values of applied voltage (5 V and 10 V), numbers of steps, and time (1 steps of 1200 s, 20 or 30 steps of 60 s each for TiN- nano/micro suspensions; 4 steps of 300 s for TiN- nano suspension) are considered as the main experimental parameters to investigate during the EPD.

The higher potential value used for suspensions containing micrometer powders were selected because of their ζ values with respect to that of the TiN-nano suspension (Table 2). Indeed, the electrophoretic mobility of the particles depends linearly on the ζ potential and it is related to the deposited yield, according to the Hamaker equation [43]. The first deposition tests were performed in *single-step* mode. Preliminary experiments performed at different times in the range from 600 s up to 3600 s suggest that 1200 s is the minimum value of time required to reduce the defects in Ti porous substrates. Figure 6 shows representative SEM micrographs of the deposits obtained after 1200 s (named EPD TiN-01, EPD TiN-02, and EPD TiN-03). From the insets of all micrographs the presence of TiN powders on the substrate surfaces, especially within the open porosity is evident. In all three cases, the surface defects were clearly reduced and partially covered with an improvement of the topology.

The infiltration of TiN powders in the pores of Ti substrates was also confirmed by the EDS maps (Figure 7).

However, the *single-step* procedure did not allow the complete coverage of pores and surface defects, even when prolonged deposition times were applied. Although longer deposition times lead to a better result, SEM micrographs of EPD TiN-04 and EPD TiN-05 deposits (not reported here) still show a partial coverage and the presence of open craters even after 3600 s of deposition time. Hence, the *single-step* method has not proved to be effective in fully optimizing the surface morphology of substrates. Therefore, the use of a *multi-step* process was then considered. In this regard, depositions were carried out in succession by varying the applied potential and the duration of the steps depending on the particle size of the suspended powders. The multi-step experiments (EPD TiN06, EPD TiN07, EPD TiN08 and EPD TiN09, see Table 1) were designed to investigate the relationship of particle size on the reduction of roughness and defects in Ti porous substrates. In Figure 8 SEM micrographs of obtained deposits are shown. The performed experiments suggest that this operating procedure, especially the sequential deposition of powders of micrometric and nanometric dimensions, allowed a more effective coverage of defects and surface porosities, improving greatly the surface quality of the porous substrate: even the larger and deeper craters result covered. On the contrary, the use of nanometer-sized powders (deposit termed EPD TiN-09) does not allow adequate coverage even in *multi-step* mode with a deposition time of 2400 s, probably because of the inadequacy of too small powders to effectively fill larger defects. The powders characterized by a single distribution of dimensions, even nanometric, probably tend to pack inside the defect, creating metastable bridges that prevent the particles from further settling movements and hinder the complete filling and covering of the crater/pore. Similarly to what happens in powder metallurgy, it appears convenient to add a coarser fraction of powders to the finer fraction (bimodal distribution), as to ensure an effective filling of the void volume. In fact, the finer particles can occupy the interstices left by the coarser particles [73]. Finally, a more prolonged deposition time up to 1800 s using suspensions containing TiN-micro sized powders (EPD TiN-08) seemed fully effective. However, this increase in deposition time using TiN micro powders followed by a deposition of 1200 s of nano particles gave a reduction of defects/roughness such as those showed by EPD TiN-06 and EPD TiN-07, where a shorter total deposition time of 2400 s was used. Therefore, 2400 s can be considered as the optimal deposition time for the multi-step mode.

### 3.5. Surface Topology Characterizations

In order to study the surface modifications occurred, a surface analysis of some prepared samples was performed by stylus contact profilometry, applied to both substrates and treated samples. Moreover, the surface topology of the EPD treated samples was investigated and the residual porosity evaluated by means of image analysis and statistical data treatment (Figure 9), starting from the optical microscopy characterization (Figure 10). Table 3 summarizes the results of the roughness analyses.

For all the samples a roughness Ra around 0.3 µm is observed, except for the sample EPD TiN-08, which shows a reduction of about 50% compared to the bare polished titanium substrate. The Rq parameter provides simple statistical handling and allows stable results as the parameter is not significantly scratched and contaminated. It was found that the samples EPD TiN-08 and EPD TiN-06 show the lowest values of surface roughness, whereas EPD TiN-09 shows to be very similar to the untreated substrate. Therefore, it is evident that a prolonged deposition treatment of TiN powders, with a higher particle size or at least a bimodal size distribution, is more effective in reducing both Ra and Rq values. Furthermore, as observed in the SEM micrographs of Figure 10, the deposition of nanometric powders alone even for prolonged times (EPD TiN-09) is not effective in covering the surface defects, nor in reducing the roughness of Ti porous substrate. Rsk identifies the symmetry differences on profiles having the same Ra or Rq value. In all the samples, negative skewness values are observed, precisely because the material is porous, and the surface profile is consequently characterized by notable and numerous valleys and very few peaks. In this case, the EPD TiN-06 and EPD TiN-08 samples prove to be those in which the treatment has reduced the depth and number of valleys the most, with a probable reduction in the average size of the pores. On the contrary, this behavior was not observed for the sample EPD TiN-09. Rku is a measure of the sharpness of the profile. It refers to the geometry of the tip of peaks and valleys. Unlike the Rsk parameter, kurtosis not only detects if the profile spikes are evenly distributed but it also measures the spikiness of the area. A spiky surface exhibits high kurtosis value and a bumpy surface presents a low kurtosis value. In all the samples, the kurtosis value was very high due to the significant sharpness of the valleys that make up the porosity in the material. The EPD TiN-06 sample, followed by EPD TiN-08, shows to be the one with the least sharp porosity, or with the most rounded profile, while the EPD TiN-09 sample shows the highest value of kurtosis, with very sharp and strongly conical valleys. Considering all the investigated parameters, the EPD TiN-09 sample is the most similar in terms of geometry to the untreated titanium substrate. On the other hand, EPD TiN-07, while remaining at the same level of average roughness of the substrate, shows an important modification of the geometry of the valleys after the EPD treatment. Beside the topological characterization of treated samples, the image analysis on optical microscopy pictures has been carried out to evaluate the overall residual porosity after EPD treatments. Measurements were performed on five pictures collected on five different areas of each sample to better characterize the state of the whole surface. Then, a statistical analysis on measured data was carried out and corresponding frequency histograms were constructed to visualize residual pore sizes of different treated surfaces, as visible in Figure 9.

Pores’ area and Feret’s diameter are reported respectively in Figure 9a,b. The frequency of a particular value is the number of times the value occurs in the data set. In this case the data set is the totality of observed pores on each sample surface. The percentage frequency distribution shows the percentage of observations falling in each class interval. Therefore, pores’ area and Feret’s diameter were grouped into categories (or class intervals) and the measured pores from microscopic pictures were assigned to different categories on the basis of their evaluated dimensions. From values of percentage frequency, it can be observed that both the area and the Feret’s diameter of pores are respectively centered at 200–400 µm^2^ and 30–50 µm for all samples. As expected, the frequency progressively decreases as the parameter value increases with trends that appear slightly different among samples. The frequency distribution of sample EPD TiN-09 follows that of Ti substrate for both pores’ area and Feret’s diameter, indicating that unsatisfying coverage of porosity has occurred after EPD treatment. The two distributions, i.e., Ti and EPD TiN-09, are centered on 300 µm^2^ and 40 µm respectively and are quite wider than those of other samples. In particular, EPD TiN-06 and EPD TiN-08 exhibit the highest concentration of smallest residual pores. In fact, they show higher frequencies of both pores’ area and Feret’s diameter at lower values, as 200 µm^2^ and 30 µm, indicating that smaller pores are detected on those surfaces, and that most of the pores are smaller than on untreated Ti surface. On the contrary, EPD TiN-07 and EPD TiN-09 show a lower frequency in the same categories, revealing higher concentration of pores having area over 300 µm^2^. For values of pores’ area over 400 µm^2^, sample EPD TiN-09 shows the highest frequency. It also represents the surface with the lowest frequency of smaller pores with respect to other samples. This means that residual porosity in sample EPD TiN-09 is quite larger than other treated surfaces, revealing a less efficient coverage of the native pores with the EPD treatment carried out with only nanometric powders. The best performing surfaces in terms of area of residual pores are EPD TiN-08 and EPD TiN-06, for which most of the pores detected are concentrated in the first two categories (smaller dimensions). This count progressively decreases in the categories relating to the larger dimensions taken into consideration (i.e., over 600 µm^2^). Moreover, the Feret’s diameter is centered for sample EPD TiN-08 in the first categories, exhibiting the highest frequency for values below 40 µm. The Feret’s diameter of pores on sample EPD TiN-09 is larger than other surfaces, exhibiting higher frequencies for values over 50 µm. Therefore, the mixed blend nano/micro or the prolonged use of microparticles seem to be good routes to improve the surface features of the substrates. As already observed from other topological results, EPD TiN-07 exhibits an intermediate behavior, between the most promising results of EPD TiN-08 and EPD TiN-06, and the poorest of EPD TiN-09. To confirm these evaluations optical microscope pictures are reported in Figure 10, where treated surfaces are compared to the same untreated state.

On each picture of Figure 10, the right side corresponds to the untreated Ti substrate, while the left one is the EPD-treated side. On the untreated side, the initial porosity of the substrate after polishing is easily visible. It can be clearly observed that sample EPD TiN-09 exhibits the worst coverage, compared to other treated surfaces, since numerous residual pores can be detected, while other samples show a good filling of the initial porosity. This confirms the results obtained after profilometric data elaborations and microscopic investigation.

## 4. Conclusions

In this work, the EPD infiltration of TiN powders into commercial porous Ti discs was evaluated for improving their surface properties. To this purpose, environmentally eco-friendly aqueous suspensions of TiN powders with nanometer and micrometer grain dimensions were investigated. A *home-made* EPD cell was specifically designed to allow the infiltration of TiN powders only on one side of the substrate. The influence of various experimental parameters (the applied voltage, duration and grain dimensions of TiN powders) was evaluated on surface morphology and roughness of TiN-infiltrated substrates. Moreover, two different procedures were investigated: the *single-step* process and the *multi-step* procedure based on *step-by-step* subsequent stages. The *single-step* process allows an improvement of the surface topology, but not the complete coverage of pores and surface defects, even at prolonged deposition times. On the other hand, the *multi-step* procedure is more effective for significantly improving the surface morphology and decreasing the roughness of Ti porous surface. The mixed blend of nano and microparticles and/or the prolonged use of microparticles reveal to be good routes to effectively optimize the surface features of the substrates. In particular, the use of powders with bimodal grain dimensions seems convenient to ensure an effective filling of the void volume: indeed, the finer particles can occupy the interstices left by the coarser particles. On the contrary, the use of nanometer-sized powders alone does not allow adequate coverage even in *multi-step* mode, probably because of the inadequacy of too small powders to effectively fill larger defects. To conclude, the results of the designed experiments suggest that the EPD infiltration of TiN powders based on a *multi-step* procedure is an effective method for the reduction of the surface defects of commercial porous Ti discs. It is important to remark that EPD has been used for the first time for infiltration of TiN powders to obtain suitable porous Ti supports. The optimized substrates covered by homogeneous TiNx-based coatings can be considered promising hydrogen separation and purification membranes. The EPD infiltrated Ti substrates display more homogenous and smaller residual pores than the commercially available Ti porous substrates, characterized by relatively irregular large pores with a wide pore size distribution and very rough surface. Therefore, the EPD infiltration makes Ti substrates suitable metallic supports for the PVD deposition of dense and defect-free TiNx.

## Figures and Tables

**Figure 1 membranes-12-00531-f001:**
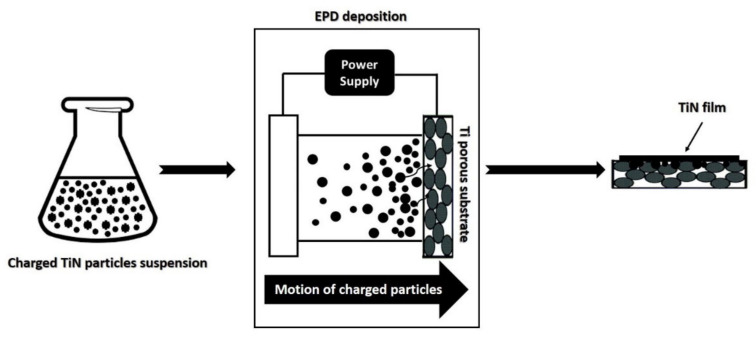
Flow chart of the EPD deposition process.

**Figure 2 membranes-12-00531-f002:**
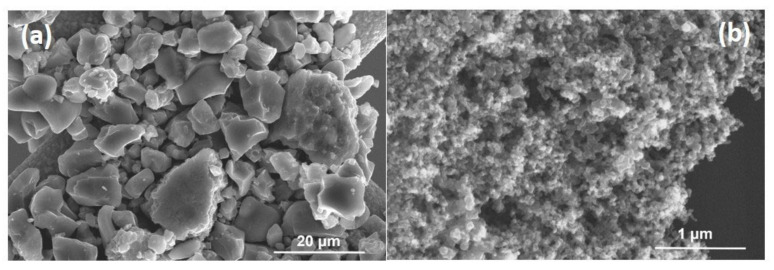
SEM micrographs of as-received micro-TiN (**a**) and nano-TiN (**b**) powders.

**Figure 3 membranes-12-00531-f003:**
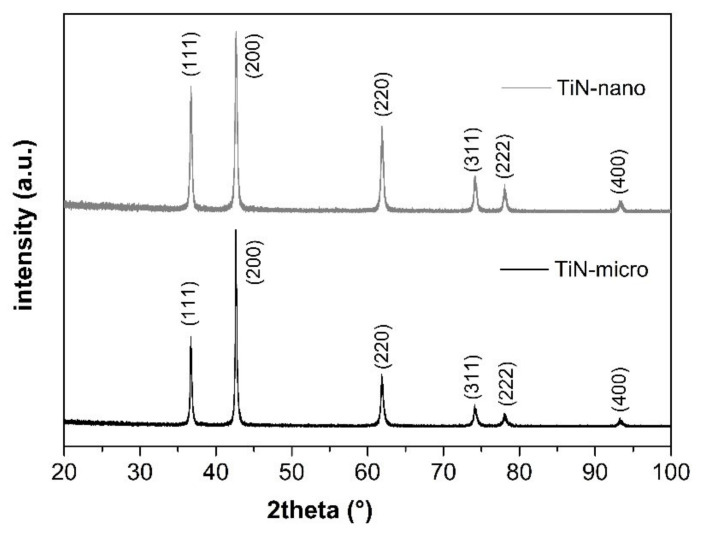
XRD patterns of as received micro-TiN e nano-TiN powders.

**Figure 4 membranes-12-00531-f004:**
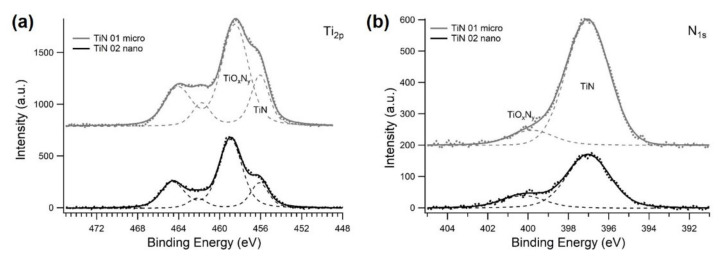
(**a**) Ti 2p XPS spectra of as-received micro- (grey color) and nano-TiN (black color) powders. The spectra are shown decomposed into Voigt-doublets (dash curves), according to the spin-orbit splitting of Ti 2p levels. (**b**) N1s spectra of as-received micro- (grey color) and nano-TiN (black color) powders. Dash curves show the deconvolution of spectra.

**Figure 5 membranes-12-00531-f005:**
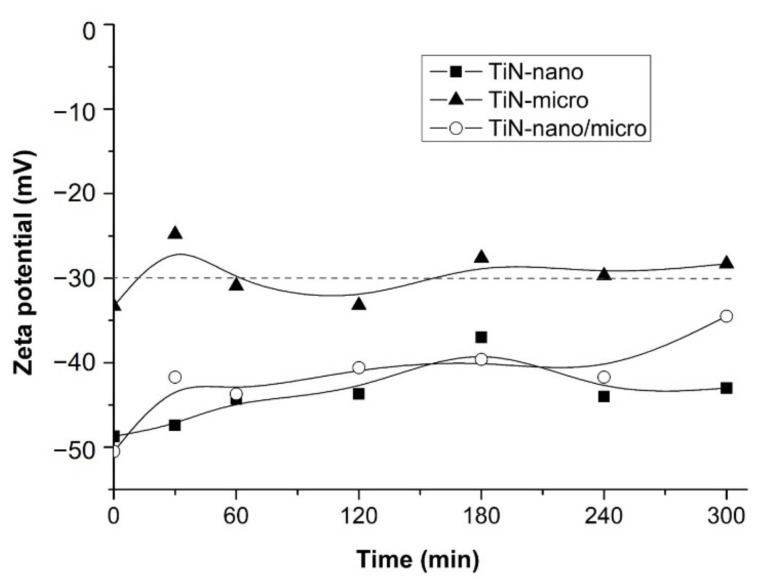
ζ potential stability of TiN-nano, TiN-micro, and TiN-nano/micro aqueous suspensions during time. The dash line represents the stability limit.

**Figure 6 membranes-12-00531-f006:**
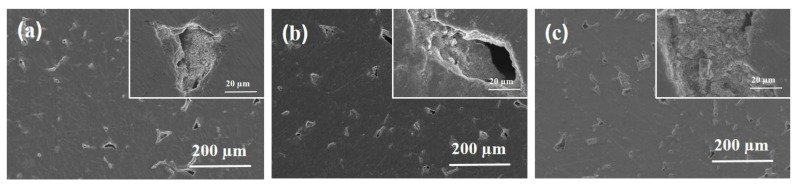
SEM micrographs of the deposits obtained on porous Ti substrates by the single-step procedure starting from suspensions of TiN-micro (EPD TiN-01 (**a**), TiN-nano (EPD TiN-02 (**b**)) and TiN-nano/micro (EPD TiN-03, (**c**) powders after 1200 s of deposition time. The respective high magnification images are shown within the inset of the images.

**Figure 7 membranes-12-00531-f007:**
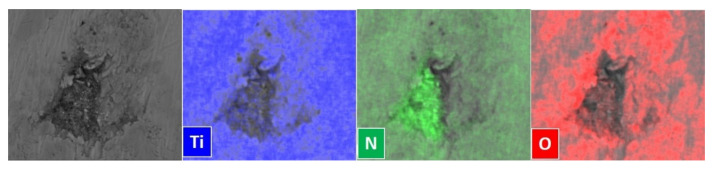
SEM micrograph and EDS maps of EPD TiN-03 sample obtained after EPD deposition of TiN nano/micro powders on a porous Ti substrate.

**Figure 8 membranes-12-00531-f008:**
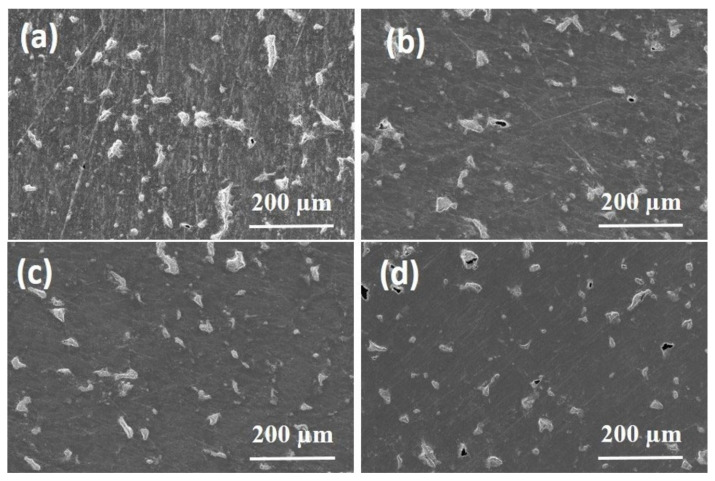
SEM micrographs of the deposits obtained on porous Ti substrates by the *multi-step* procedure: (**a**) EPD TiN-06, (**b**) EPD TiN-07, (**c**) EPD TiN-08, (**d**) EPD TiN-09).

**Figure 9 membranes-12-00531-f009:**
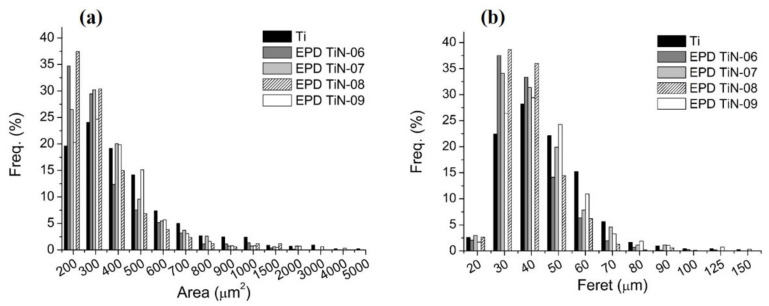
Statistical analysis performed on microscope images to evaluate residual porosity after EPD treatments. Frequent distribution of (**a**) pores’ area and (**b**) pores’ Feret diameter.

**Figure 10 membranes-12-00531-f010:**
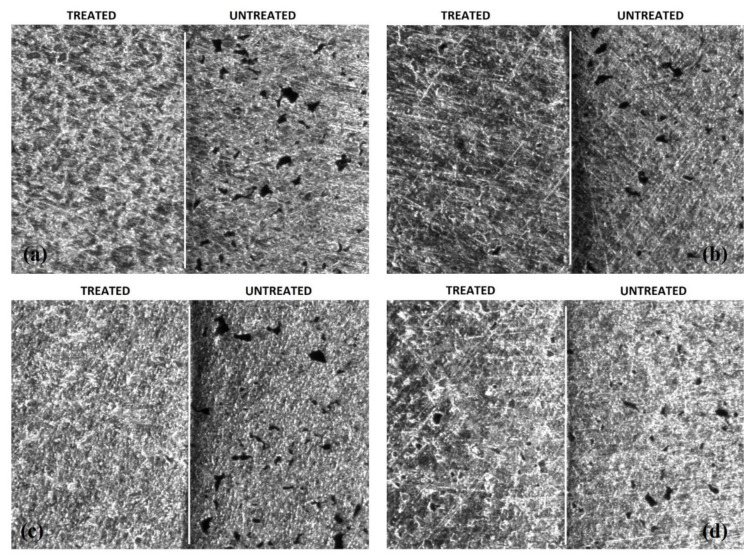
Optical microscope pictures (40x) of treated/untreated surfaces of (**a**) EPD TiN-06, (**b**) EPD TiN-07, (**c**) EPD TiN-08, (**d**) EPD TiN-09.

**Table 1 membranes-12-00531-t001:** Selection of EPD experiments.

Sample	TiN Suspension (1 wt%/)	Potential (V)	Number of Steps	Step Duration (s)	Duration (s)
** *Single-step* **					
**EPD TiN-01**	TiN-micro	10	1	\	1200
**EPD TiN-02**	TiN-nano	5	1	\	1200
**EPD TiN-03**	TiN-nano/micro	10	1	\	1200
**EPD TiN-04**	TiN-nano/micro	10	1	\	3600
**EPD TiN-05**	TiN-nano	5	1	\	3600
** *Multi-step* **					
**EPD TiN-06**	TiN-nano/micro TiN-nano	10 5	20 4	60 300	1200 1200
**EPD TiN-07**	TiN-micro TiN-nano	10 5	30 4	60 300	1800 1200
**EPD TiN-08**	TiN-micro TiN-nano	10 5	20 4	60 300	1200 1200
**EPD TiN-09**	TiN-nano	5	8	300	2400

**Table 2 membranes-12-00531-t002:** ζ potential of TiN-nano, TiN-micro, and TiN-nano/micro suspensions obtained in aqueous media.

Sample	ζ Potential (mV)
**TiN-nano**	−50.5 ± 3.7
**TiN-micro**	−33.3 ± 3.1
**TiN-nano/micro**	−46.7 ± 4.8

**Table 3 membranes-12-00531-t003:** Surface roughness parameters obtained from line profiles for Ti polished and infiltrated substrates.

Sample	Ra	Rq	Rsk	Rku
**EPD TiN-06**	0.309 ± 0.059	0.573 ± 0.170	−2.916 ± 0.782	17.443 ± 5.07
**EPD TiN-07**	0.366 ± 0.081	0.669 ± 0.189	−3.514 ± 0.323	19.900 ± 2.729
**EPD TiN-08**	0.179 ± 0.356	0.389 ± 0.110	−3.079 ± 0.451	19.317 ± 2.052
**EPD TiN-09**	0.399 ± 0.035	0.865 ± 0.080	−4.829 ± 0.461	31.272 ± 6.086
**Ti**	0.376 ± 0.053	0.803 ± 0.137	−4.888 ± 0.741	34.193 ± 7.943

## Data Availability

Data are available on request due to privacy restrictions. The data presented in this study are available on request from the corresponding author.

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
