# Peer review of "Surface Optimization of Commercial Porous Ti Substrates by EPD of Titanium Nitride"

_membranes, 2022, doi:10.3390/membranes12050531_

Round 1

Reviewer 1 Report

​Comments to the authors

Abstract: the introduction of materials and method (lines 21-27) is much larger than the work done in the present work. Concise it in 2-3 lines. 

Introduction: A concise introduction of the pros and cons of EPD technology over other techniques (like direct current deposition, electrochemical deposition etc) can be beneficial to the readers. Some literature can be cited for this purpose:

  1. Electrophoretic Deposition (EPD): Fundamentals and Applications from Nano- to Microscale Structures, Handbook of Nanoelectrochemistry pp 561-591 https://doi.org/10.1007/978-3-319-15266-0_7.
  2. Progress in electrochemical and electrophoretic deposition of nickel with carbonaceous allotropes: a review, Advanced Materials Interfaces, https://doi.org/10.1002/admi.201901096
  3. A review on fundamentals and applicationsof electrophoretic deposition (EPD), Progress in Materials Science, 52 (2007) 1–61.

Results and discussion: In XRD the phases present for all characteristic peaks should be mentioned in the text as well as in figure 2. 

Mechanical polishing is very generic for substrate preparation. It must be removed from the paper. No need of Figures 4 and 5. 

In figure 8, the respective high mag. images can be given within the ​​inset of the images. As image b can be given inside image a..and same for others too. 

Why somewhere it is mentioned that the coatings were deposited using PVD technique? It is confusing. 

The only thing which is clear that the multi-process is better than the single step. Why authors made so many samples and vary so many parameters? The results and discussion didn't explain well about all these samples and parameters. 

Readers have to check the table again to understand the sample composition and parameters varied for those. There is no organized way to prepare samples and to vary the parameters for them. Somewhere time was used 1200 s, somewhere 1800/3600/2400..same for potential and step number. One can not read the paper without looking at table 1. 

There is no conclusion showing the effect of varying size, time, and potential on different coatings. 

Finally, the paper is not up to the mark for this Journal. 

Author Response

we really appreciate the reviewer’s comments related to our manuscript entitled “Surface optimization of commercial porous Ti substrates by EPD of titanium nitride”. The corresponding response to each comment is as following, while the modifications are highlighted in yellow color in the revised manuscript.

Best regards,

Maria Cannio

Reviewer 2 Report

The authors have nicely presented the surface optimization of porous Ti Substrates by electrophoretic deposition (EPD) of titanium nitride. Various aspects of current PVD based technique is well explained with the limitations and its effects on deposited layers. The novel approach by EPD to overcome the effects of conventional techniques is reported satisfactorily.  The influence of various deposition parameters on surface morphology and roughness of TiN-infiltrated substrates were investigated in the study. The results presented suggest a multi-step EPD procedure as a successful technique to reduce the substrate surface defects of porous Ti substrates. However broader range of techniques of nano powder processing in porous medium is to be discussed.

Suggestion: It is necessary to explain broader range of nanopowder processing through porous medium.

The following article on one of the applications of nanoparticle seeded resin on formation of meso and micro fractals can be discussed in section “Introduction” in the context of giving broader range of similar processes.

  1. Ajinkya Anil Singare, B. S. Kale and Kiran Suresh Bhole, "Experimental Characterization of Meso-Micro Fractals from Nanoparticle Seeded Resin in Lifting Plate Hele-Shaw Cell," Materials Today Proceedings, 2018, Vol. 5, issue 11, part3, pp. 24213-24220, doi: https://doi.org/10.1016/j.matpr.2018.10.216

Author Response

we really appreciate the reviewer’s comments related to our manuscript entitled “Surface optimization of commercial porous Ti substrates by EPD of titanium nitride”. The corresponding response to each comment is as following, while the modifications are highlighted in yellow color in the revised manuscript.

Best regards,

Dr. M. Cannio

Reviewer 3 Report

Comments: This paper reports the Surface optimization of commercial porous Ti substrates by EPD of titanium nitride. Authors should address the following minor comments for its acceptance.

  1. The conclusion should be rewritten with novelty, limitations, and the importance of the investigation in the present study
  2. The author need to index peaks position in the XRD patterns of micro-TiN and nano-TiN powders. Also check the figure caption (Figure 2).
  3. Author needs to supply the XPS survey spectra of micro-TiN and nano-TiN to determine their elemental composition.
  4. In order to show the superiority of the current materials, comparisons over the other related materials reported in the past works of literature are necessary. Surface properties of the current materials have to be compared with those of the other materials, and performance improvements have to be discussed.
  5. There are some mistakes in the manuscript. The authors need to check and double-check the whole manuscript to get rid of some syntax and format errors.

Author Response

(The authors gave the same response as above.)

Round 2

Reviewer 1 Report

Though the suggested reference 2 is based on Ni-carbonaceous coatings, but the article includes a detailed introduction of Electrophoretic deposition. 

The explanation of EPD-TibN06, 07, 08, 09 can be given in single paragraph. Remove short paragraphs for each sample . Merge them in single paragraph. 

Author Response

Dear reviever,

thank you for your useful comments. I provide a point-by-point response to them as following: 

1) English language and style are fine/minor spell check required

We revised the paper and corrected some mistakes

2) Though the suggested reference 2 is based on Ni-carbonaceous coatings, but the article includes a detailed introduction of Electrophoretic deposition. 

Done. We added this reference in line 113, where EPD properties are described.

3) The explanation of EPD-TibN06, 07, 08, 09 can be given in single paragraph. Remove short paragraphs for each sample . Merge them in single paragraph. 

Done. We merged the different explanations for the prepartion of EPD-TiN06, 07, 08, 09 samples in a single paragraph
